# The Inflammatory Nexus: Unraveling Shared Pathways and Promising Treatments in Alzheimer’s Disease and Schizophrenia

**DOI:** 10.3390/ijms26136237

**Published:** 2025-06-27

**Authors:** Aurelio Pio Russo, Ylenia Pastorello, Lóránd Dénes, Klara Brînzaniuc, Jerzy Krupinski, Mark Slevin

**Affiliations:** 1Department of Anatomy and Embryology, George Emil Palade University of Medicine, Pharmacy, Science, and Technology of Târgu Mureş, 540139 Târgu Mureş, Romania; aurelio.russo@umfst.ro (A.P.R.); ylenia.pastorello@gmail.com (Y.P.); lorand.denes@umfst.ro (L.D.); klara.brinzaniuc@umfst.ro (K.B.); 2Doctoral School of Medicine and Pharmacy, George Emil Palade University of Medicine, Pharmacy, Science, and Technology of Târgu Mureş, 540139 Târgu Mureş, Romania; 3Department of Neurology, MútuaTerrassa University Hospital, 08221 Barcelona, Spain; jkrupinski@mutuaterrassa.es; 4Department of Life Sciences, Manchester Metropolitan University, Manchester M1 5GD, UK; 5Centre for Advanced Medical and Pharmaceutical Research, George Emil Palade University of Medicine, Pharmacy, Science, and Technology of Târgu Mureş, 540139 Târgu Mureş, Romania

**Keywords:** Alzheimer’s disease, schizophrenia, S100 calcium binding protein B (S100B), neuroinflammation, receptor for advanced glycation end-products (RAGE)

## Abstract

Alzheimer’s disease (AD) and schizophrenia are traditionally considered distinct clinical entities, yet growing evidence highlights substantial overlap in their molecular and neuroinflammatory pathogenesis. This review explores current insights into the shared and divergent mechanisms underlying these disorders, with emphasis on neuroinflammation, autophagy dysfunction, blood–brain barrier (BBB) disruption, and cognitive impairment. We examine key signaling pathways, particularly spleen tyrosine kinase (SYK), the mechanistic (or mammalian) target of rapamycin (mTOR), and the S100 calcium-binding protein B (S100B)/receptor for advanced glycation end-products (RAGE) axis, that link glial activation, excitatory/inhibitory neurotransmitter imbalances, and impaired proteostasis across both disorders. Specific biomarkers such as S100B, matrix metalloproteinase 9 (MMP9), and soluble RAGE show promise for stratifying disease subtypes and predicting treatment response. Moreover, psychiatric symptoms frequently precede cognitive decline in both AD and schizophrenia, suggesting that mood and behavioral disturbances may serve as early diagnostic indicators. The roles of autophagic failure, cellular senescence, and impaired glymphatic clearance are also explored as contributors to chronic inflammation and neurodegeneration. Current treatments, including cholinesterase inhibitors and antipsychotics, primarily offer symptomatic relief, while emerging therapeutic approaches target upstream molecular drivers, such as mTOR inhibition and RAGE antagonism. Finally, we discuss the future potential of personalized medicine guided by genetic, neuroimaging, and biomarker profiles to optimize diagnosis and treatment strategies in both AD and schizophrenia. A greater understanding of the pathophysiological convergence between these disorders may pave the way for cross-diagnostic interventions and improved clinical outcomes.

## 1. Introduction

Neurodegenerative Disorders (NDDs) are disorders involving progressive neuronal dysfunction and death, leading to cognitive, motor, and functional disorders [1]. The most common NDDs include Alzheimer’s Disease (AD), Parkinson’s Disease (PD), Amyotrophic Lateral Sclerosis (ALS), and Multiple Sclerosis (MS). Overall, these diseases are affecting 15% of the global population [1,2].

Although these disorders vary in clinical manifestations and pathophysiological factors, they share key pathological mechanisms. A prominent feature is the buildup of misfolded proteins, which interferes with critical cellular functions like autophagy, mitochondrial regulation, and calcium balance, thereby accelerating neuronal degeneration [2,3]. In AD specifically, neuronal function is significantly impaired by the presence of amyloid-beta (Aβ) plaques and tau protein tangles [2]. Another common hallmark is region-specific neuronal susceptibility—for instance, the hippocampus in AD and the substantia nigra in PD, determined by both genetic predispositions and environmental influences [3]. AD almost always appears later in life, with the majority of cases diagnosed after age 65, although rare early-onset forms can begin as early as 30–60 years old. The typical onset for AD is around 50–60 years or older [4,5].

While many NDDs occur sporadically, hereditary cases have provided critical insights into disease biology. Mutations in presenilin 1 (PSEN1), presenilin 2 (PSEN2), and amyloid precursor protein (APP) are associated with early-onset AD, whereas familial PD has been linked to genetic variants in leucine-rich repeat kinase 2 (LRRK2) and synuclein alpha (SNCA) [3,6]. Furthermore, exposure to neurotoxic substances and incidents such as traumatic brain injury (TBI) have been identified as risk-enhancing factors, underscoring the intricate interplay between genetic and environmental factors [3,6].

Neuroinflammation, driven by microglia and astrocyte activation, is increasingly recognized as a key factor in NDD pathogenesis. While acute inflammation is protective in the first stages, chronic inflammation promotes neuronal dysfunction, cell death, and protein aggregation, accelerating disease progression [7].

Psychiatric disorders are increasingly recognized as both comorbidities and potential early indicators of neurodegenerative disorders. For example, late depression and anxiety have been associated with a higher risk of developing AD, suggesting shared neuropathological mechanisms such as hippocampal atrophy, dysregulation of the hypothalamic-pituitary–adrenal axis, and chronic inflammation [8]. Additionally, behavioral and psychiatric symptoms such as apathy, psychosis, and agitation are frequently observed in NDDs, especially in AD, complicating both diagnosis and management.

Individuals with AD are prone to depression; research shows that about 32.3% of AD patients have depressive symptoms [9]. Reduced quality of life and accelerated cognitive decline are linked to depression in AD patients. Furthermore, a history of depression has been linked to an increased risk of developing AD in later life, indicating a reciprocal relationship between the two disorders [10]. Patients with AD are also frequently found to have anxiety disorders. According to recent studies, anxiety symptoms can worsen cognitive impairment and lead to behavioral abnormalities, being also linked to an increased risk of developing AD [11,12,13]. Dementia and AD are more likely to develop in people with schizophrenia. According to a meta-analysis with more than 5 million participants, people with schizophrenia have a 2.29 relative risk of dementia, which is much higher than the risk for people without schizophrenia [12]. It is worth mentioning that schizophrenia, contrarily to AD, typically manifests in late adolescence or early adulthood, most commonly between the ages of 10 and 30, with rare cases before the age of 12 or after 40 [13,14]. Common pathophysiological mechanisms, including neuroinflammation, neurotransmitter imbalances, and vascular alterations, may be responsible for the co-occurrence of AD and psychiatric disorders. Chronic inflammation, for example, has been linked to both AD and depression, indicating that inflammatory processes may be a common factor in both [10]. The presence of psychiatric comorbidities in AD patients poses significant challenges for diagnosis and treatment [15]. These comorbidities can mask or mimic AD symptoms, leading to delayed diagnoses. Additionally, mental health issues can worsen cognitive impairments, decrease treatment compliance, and consequently worsen the overall condition [10]. Therefore, in this narrative review, we will comprehensively assess the molecular and neuroinflammatory crosstalk between AD and schizophrenia, with a focus on their shared pathophysiological mechanisms, potential biomarkers, and emerging therapeutic targets.

## 2. Neuropathological Processes in AD and Schizophrenia

### 2.1. Molecular and Morphological Signatures

Schizophrenia is a chronic psychiatric disorder characterized by symptoms such as hallucinations, delusions, and cognitive impairments [16]. AD is a neurodegenerative disease marked by progressive memory loss and cognitive decline [17]. According to epidemiological research, people with schizophrenia are more than twice as likely to develop dementia, including AD, in later life [18,19]. Genetic studies indicate connections between AD and schizophrenia. Genetically common variants thought to be involved in the pathophysiology of both disorders have recently been discovered by genome-wide association studies. For instance, a study revealed a strong genetic correlation between AD and schizophrenia, suggesting that shared genetic risk factors could influence both disorders [20]. Even though some recent studies have suggested a statistically significant polygenic overlap between schizophrenia and AD, this overlap is limited. In detail, genome-wide association studies (GWAS) have demonstrated that both conditions have largely distinct sets of susceptibility genes, although both are highly heritable disorders. For AD, GWAS have identified dozens of risk loci, including genes involved in microglial function, immune response, and protein catabolism, such as apolipoprotein E (APOE), clusterin (CLU), phosphatidylinositol binding clathrin assembly protein (PICALM), complement receptor 1 (CR1), bridging integrator 1 (BR1), adenosine tri-phosphate binding cassette subfamily A member 7 (ABCA7), membrane spanning 4-domains A (MS4A), cluster of differentiation 2 associated protein (CD2AP), and ephrin type-A receptor 1 (EPHA1) [21,22,23]. In contrast, schizophrenia GWAS have identified hundreds of risk loci, with a strong emphasis on genes related to neuronal and synaptic function, including neurexin 1 (NRXN1), calcium voltage-gated channel subunit alpha1 C (CACNA1C), and glutamate ionotropic receptor N-methyl-D-aspartate type subunit 2A (GRIN2A) [24,25]. The largest GWAS for AD identified 38 risk loci, while the largest for schizophrenia reported 287 risk loci, and only one genome-wide significant single-nucleotide polymorphism was found to be shared between the two conditions [26,27]. This indicates that, despite some shared biological pathways and a degree of genetic correlation, the susceptibility genes identified by GWAS for AD and schizophrenia seem to be largely distinct. 

Dysfunctions in the cholinergic system, which is essential to cognitive functions, are linked to both AD and schizophrenia [28]. In AD, there is a severe loss of cholinergic neurons in the basal forebrain, specifically within the nucleus basalis. This leads to a significant reduction in cortical and hippocampal cholinergic supply [29,30]. Schizophrenia is also associated with a dysfunction of the cholinergic neurons in the basal forebrain. Studies involving the use of in vivo brain magnetic resonance imaging (MRI) in individuals with schizophrenia and post-mortem assessments of patients with schizophrenia show reduced dimension and availability of both muscarinic and nicotinic receptors in key brain regions involved in cognitive functions, such as the hippocampus and prefrontal cortex [31,32]. Treatments that target muscarinic receptors have been developed because of this common cholinergic dysfunction. Notably, the first new mechanism in more than 50 years was introduced when the Food and Drug Administration (FDA) approved xanomeline-trospium, a novel treatment for schizophrenia [33]. Aβ plaques and tau neurofibrillary tangles are hallmark pathologies of AD. As reviewed by Eickhoff et al., emerging evidence obtained from post-mortem studies, in vivo molecular imaging studies, and structural brain MRI studies suggests that similar amyloid and tau pathologies may also be present in the brains of individuals with schizophrenia, although to a lesser extent [34]. These findings point to a specific molecular background shared by AD and schizophrenia, suggesting that impairment of the autophagy pathway regulation system may contribute to both disorders [35]. Lastly, neuroinflammation is a common feature in both disorders. Microglial activation, indicative of an inflammatory response in the brain, has been observed in both conditions. This shared inflammatory behavior suggests that neuroinflammation may play a role in the pathophysiology of both disorders [36,37,38].

Glutamatergic signaling, mediated by glutamate, is a key factor for normal brain functioning, including synaptic transmission, plasticity, learning, and memory. Disruptions of this system have been implicated in both schizophrenia and AD, suggesting shared pathophysiological mechanisms [39]. In schizophrenia, dysfunctions in glutamatergic neurotransmission have been identified, particularly involving N-methyl-D-aspartate (NMDA) receptors. Hypofunction of these may contribute to cognitive deficits and negative symptoms observed in subjects with schizophrenia [40]. In AD, glutamatergic dysfunction is associated with excitotoxicity, where the excessive activation of NMDA receptors leads to neuronal injury and death. This toxicity is associated with Aβ accumulation, which can disrupt glutamate uptake and enhance receptor activation, contributing to cognitive decline [41]. Both disorders exhibit alterations in glutamate transporters and receptor subunits, leading to impaired synaptic function. Additionally, oxidative stress and mitochondrial dysfunction can exacerbate glutamatergic dysregulation, further contributing to neurodegeneration. Beyond the glutamatergic system, other molecular pathways are also deeply involved and interact with these complex neurodegenerative and psychiatric conditions. For instance, spleen tyrosine kinase (SYK) is a non-receptor tyrosine kinase playing a pivotal role in mediating immune responses through its involvement in signaling pathways activated by immunoreceptor tyrosine-based activation motifs (ITAMs) [42]. 

Microglia utilize SYK to respond to pathological stimuli, including Aβ aggregates. The activation of SYK enhances microglial phagocytic activity, facilitating the clearance of Aβ plaques. Conversely, the inhibition or genetic deletion of SYK impairs this clearance mechanism, leading to exacerbated amyloid pathology and cognitive decline in 5-familial Alzheimer’s disease mutations (5xFAD) mice and SYK mice models [43]. Upon activation by key factors, including damage-associated molecular patterns (DAMPs) and pathogen-associated molecular patterns (PAMPs), SYK starts a cascade leading to the production of pro-inflammatory cytokines, including interleukin 1-beta (IL-1β), tumor necrosis factor-alpha (TNF-α), and interleukin 6 (IL-6), and chemokines, such as the chemokine C-C motif ligand 2 (CCL2) and C-X-C motif chemokine ligand 10 (CXCL10). These, in turn, will amplify the neuroinflammatory state, which, with chronic persistence, leads to NDDs [44]. Another critical regulatory pathway implicated in these conditions is the mechanistic (or mammalian) target of rapamycin (mTOR), which is a serine/threonine kinase playing a central role in cellular progression regulation, involved in processes such as metabolism, proliferation, growth, and survival. It functions through two complexes: the mammalian target of rapamycin complex 1 (mTORC1) and the mammalian target of rapamycin complex 2 (mTORC2) [45]. In AD, impaired autophagy due to the hyperactivation of mTORC1 has been observed, leading to Aβ plaques and neurofibrillary tangles accumulation. Studies have shown that the inhibition of mTORC1 increases autophagic activity, supporting the clearance of toxic aggregates and ameliorating cognitive deficits in transgenic mouse models [46,47]. Alterations in genes related to this pathway, such as Disrupted in Schizophrenia 1 (DISC1), have been linked to schizophrenia. In this case, DISC1 interacts with the mTOR pathway, influencing neuronal development and synaptic function [48]. Analyses of post-mortem brain tissues from individuals with schizophrenia have shown reduced expression and reduced phosphorylation of ribosomal protein S6, a downstream effector of mTOR, thus indicating a potential pathway hypofunction [49,50].

“Type 3 diabetes” is a term increasingly used in the scientific literature to describe a specific form of diabetes that affects the brain, closely related to the pathogenesis of AD [51,52]. This concept arises from evidence that brain insulin resistance, a hallmark of type 2 diabetes mellitus, drives AD pathology by impairing amyloid-β clearance, promoting tau hyperphosphorylation, and exacerbating neuroinflammation [53,54]. Chronic hyperglycemia and insulin resistance in type 2 diabetes accelerate the formation of advanced glycation end products (AGEs), which bind to their receptor for advanced glycation end products (RAGE), triggering pro-inflammatory and oxidative pathways that synergistically worsen neurodegeneration [55,56]. In detail, RAGE is a multifunctional transmembrane receptor, part of the immunoglobulin superfamily, expressed on microglia, astrocytes, neurons, and endothelial cells in the central nervous system (CNS). It plays a crucial role in neuroinflammation and NDDs such as AD [7]. RAGE interacts with various ligands, including AGEs, S100 proteins, and high-mobility group box 1 (HMGB1) protein [57]. Upon ligand binding, RAGE activates intracellular signaling pathways such as nuclear factor kappa B (NF-κB), mitogen-activated protein kinase (MAPK), and Janus kinase/signal transducers and activators of transcription (JAK/STAT), leading to the transcription of pro-inflammatory cytokines and chemokines, as shown in Figure 1 [58].

This activation perpetuates a chronic inflammatory state within the CNS. In AD, RAGE plays a dual role, acting as both a transporter and activator of neuroinflammatory cascades. RAGE actively transports Aβ from the peripheral circulation into the brain, leading to increased Aβ deposition and plaque formation; at the same time, it inhibits its clearance by recalling low-density lipoprotein receptor-related protein 1 (LRP1), exacerbating amyloid toxicity [59]. Chronic activation of RAGE in endothelial cells of the blood–brain barrier (BBB) increases vascular permeability, allowing toxic substances to infiltrate the brain parenchyma and contributing to disruption of cerebrovascular homeostasis, leading to neuronal damage and cognitive decline [60,61]. RAGE antagonists and inhibitors, such as Azeliragon and 4-Chloro-N-cyclohexyl-N-(phenylmethyl)-benzamide (FPS-ZM1), have been explored in in vivo studies and clinical trials, showing potential in reducing neuroinflammation and amyloid pathology [62]. A reverse translational study conducted on schizophrenia individuals has shown that the overactivation of the MMP9/RAGE pathway can lead to redox dysregulation (imbalance between reactive oxygen species (ROS) production and antioxidants) and neuroinflammation, contributing to an imbalance between inhibitory and excitatory neurotransmitters, an imbalance which is involved in the pathophysiology of schizophrenia [63]. In detail, when oxidative stress is elevated, the activation of MMP-9 occurs, which in turn cleaves RAGE into soluble and nuclear forms. This process leads to the activation of NF-κB and the release of pro-inflammatory cytokines, perpetuating a state of neuroinflammation [63,64]. These findings have demonstrated that patients with schizophrenia, especially those in early stages, present elevated plasma levels of soluble RAGE and MMP9. These biomarkers are associated with lower prefrontal gamma-aminobutyric acid (GABA) levels, further supporting the link between MMP9/RAGE pathway activation, inhibitory and excitatory imbalances, and the clinical manifestations of schizophrenia [63,64].

### 2.2. Neuroinflammation and the Blood–Brain Barrier

Neuroinflammation is increasingly recognized as a critical factor in the progression of AD. Chronic activation of the brain’s immune components, particularly microglia and astrocytes, contributes to neuronal damage and accelerates disease pathology [65]. In AD patients, elevated levels of pro-inflammatory cytokines such as TNF-α, IL-1β, and chemokines like monocyte chemoattractant protein-1 (MCP-1) have been observed [1,7,37]. These molecules exacerbate neuronal injury and promote Aβ aggregation, leading to synaptic dysfunction. Microglia, the brain’s resident immune cells, initially act to clear Aβ deposits through phagocytosis. However, in AD, sustained activation leads to a detrimental inflammatory response, releasing neurotoxic substances that harm neurons [65,66]. Similarly, astrocytes, which support neuronal function, become reactive and release factors that contribute to neuroinflammation and impede neuronal repair mechanisms [65]. Addressing neuroinflammation presents a promising therapeutic avenue in AD. Strategies include modulating microglial activation, inhibiting pro-inflammatory cytokines, and enhancing anti-inflammatory pathways to mitigate neuronal damage and slow disease progression [66,67,68]. Beyond AD, neuroinflammation has been implicated in the pathophysiology of several psychiatric disorders. Chronic low-grade neuroinflammation is understood as a driving factor in conditions such as major depressive disorder, schizophrenia, anxiety disorders, and autism spectrum disorder [69]. Elevated levels of pro-inflammatory cytokines and activated microglia have been observed in patients with these disorders, suggesting that neuroinflammatory processes may contribute to their development and progression. For instance, in major depressive disorders, increased neuroinflammation has been linked to symptoms of anhedonia and fatigue, while in schizophrenia, neuroinflammatory mechanisms are thought to contribute to cognitive deficits and negative symptoms [69,70].

The blood–brain barrier is a selective, semi-permeable border that protects the CNS from potentially harmful substances while regulating the transport of essential molecules [71]. This barrier is primarily composed of endothelial cells connected by tight junctions, pericytes, astrocytic end-feet, and the basement membrane. The complex structure maintains CNS homeostasis by restricting the entry of pathogens, toxins, and peripheral immune cells, while permitting the passage of nutrients and removal of waste products [71,72]. In AD, BBB dysfunction has been implicated in the disease’s pathogenesis and progression. A variety of factors can induce BBB dysfunction and breakdown in AD, such as Aβ peptides, which induce endothelial dysfunction, leading to increased BBB permeability, chronic activation of microglia and astrocytes, cerebral amyloid angiopathy, and elevated oxidative stress [61,73]. Since the barrier is highly selective and its disruption occurs over time, innovative strategies to directly deliver therapies to the brain have been discovered and considered, such as focused ultrasound, which temporarily opens the BBB, allowing the targeted delivery of therapeutic agents [74]. Other studies focus instead on glucagon-like peptide-1 (GLP-1) receptor agonists, which were originally used for diabetes and are now under investigation for their potential neuroprotective mechanisms [75]. In AD mice and cell models, GLP-1 receptor agonists show neuroprotective effects by reducing neuroinflammation, oxidative stress, and neuronal apoptosis [75]. A systematic review of clinical trials in humans with AD found that GLP-1 receptor agonist therapy did not significantly alter Aβ and tau biomarkers or improve cognition. However, it was shown to have potential metabolic benefits, such as improved glucose levels [76]. Regarding the BBB, several GLP-1 receptor agonists are able to cross the barrier, thus possessing the crucial ability to exert direct effects on the central nervous system. This ability allows them to engage with brain targets to mediate neuroprotective and anti-inflammatory effects [75,77].

In schizophrenia, research indicates that BBB dysfunction may be involved in the dysregulation of the dopaminergic system. Disruption of the barrier allows peripheral immune cells to infiltrate the brain, exacerbating neuroinflammation and contributing to the neuronal abnormalities seen in schizophrenia [78]. A recent study reported increased BBB leakage in individuals with schizophrenia compared to healthy controls, underscoring the critical role of BBB integrity in maintaining CNS health and its involvement in the development and progression of neuropsychiatric disorders [79]. Among emerging contributors of BBB disruption, monomeric C-reactive protein (mCRP) has gained attention for its pro-inflammatory properties and the association with neurovascular damage. mCRP has been shown to directly interact with endothelial cells, inducing the expression of adhesion molecules like Intercellular Adhesion Molecule-1 (ICAM-1), which promote immune cell infiltration and perpetuate local inflammation. These effects lead to increased permeability of the BBB and compromise its structural integrity [80].

### 2.3. Autophagy, Senescence, and Neurodegeneration

Emerging research highlights the significant roles of impaired autophagy and cellular senescence in AD pathogenesis, particularly through their contributions to neuroinflammation. Autophagy is a cellular process responsible for degrading and recycling damaged organelles and misfolded proteins and at the same time maintaining neuronal health. In AD, autophagic dysfunction leads to the accumulation of toxic proteins, such as Aβ and hyperphosphorylated tau, which, as mentioned, are hallmark features of the disease. Autophagic impairment involves several mechanisms such as lysosomal dysfunction, which hampers the degradation of autophagic substrates, thus leading to the accumulation of Aβ and tau proteins; retromer complex deficiency, which is essential for moving proteins from endosomes to the trans-Golgi network; and LC3-Associated Endocytosis (LANDO), which is a novel pathway identified in microglia. LANDO facilitates the clearance of Aβ, meaning that its impairment leads to increased Aβ deposition [81]. According to the review by Zhang et al. on the impairment of the autophagy–lysosomal pathway in AD, numerous studies indicate that enhancing autophagic activity can lead to reduced Aβ levels and promote the degradation of pathological tau. This suggests the therapeutic potential of targeting this pathway as a strategy for AD [82]. Autophagy dysfunction has been associated with schizophrenia, where genes responsible for autophagy, such as Unc-51-like kinase 2 (ULK2) and beclin-1 (BECN1), exhibit significantly decreased expression in specific brain areas as shown in Table 1 [83]. The reduced expression of BECN1 in the hippocampus of schizophrenia models correlated with impaired autophagy initiation and synaptic dysfunction [84,85]. In AD, BECN1 deficiency accelerates Aβ and tau accumulation, while the BECN1 F121A mutation enhances autophagy and therefore reduces Aβ plaques and improves cognition in mouse models. Specifically, in knock-in mice models with F121A point mutation in BECN1, a cross of these F121A BECN1 with 5xFAD and platelet-derived growth factor promoter driving expression of human amyloid precursor protein (PDAPP) mice was used to study the effects of autophagy hyperactivation on AD pathology [86,87].

Cellular senescence refers to a state of permanent cell cycle arrest, where cells cease dividing but remain metabolically active. These cells secrete a variety of pro-inflammatory cytokines, chemokines, and proteases, collectively known as the senescence-associated secretory phenotype (SASP) [88]. In the context of AD, senescent cells have been identified in various brain regions, contributing to a generalized chronic inflammatory environment, thus exacerbating neurodegeneration [17].

Impaired autophagy and cellular senescence are two interconnected processes that collectively contribute to neuroinflammation: defective autophagy can lead to the accumulation of damaged cellular components, triggering cellular stress responses and promoting senescence. Senescent cells, in turn, will secrete SASP factors that will inevitably lead to autophagic process impairment, thus creating a vicious circle that exacerbates inflammation and neurodegeneration [89].

Targeting these interconnected pathways may offer potential therapeutic solutions for AD. Strategies aimed at enhancing autophagy may facilitate the clearance of toxic protein aggregates, while interventions aiming to destroy senescent cells or suppress their inflammatory secretions could mitigate chronic neuroinflammation.

**Table 1 ijms-26-06237-t001:** Comparison of how autophagy dysfunction and related molecular pathways present in AD and schizophrenia, highlighting both shared mechanisms and disease-specific outcomes. E/I imbalance: excitatory/inhibitory imbalance; PVI: parvalbumin-positive interneuron; PNN: perineuronal nets; BECN1: beclin-1; ULK2: Unc-51-like kinase 2; BCL2: B-cell lymphoma 2; ATG5: autophagy protein 5; ADNP: activity-dependent neuroprotective protein; Aβ: amyloid beta; MMP9: matrix metalloproteinase 9; RAGE: receptor for advanced glycation end-products; NF-κB: nuclear factor-kappa B; ↓: decreased; ↑: increased.

Feature	Alzheimer Syndrome	Schizophrenia Syndrome
**Main Pathology**	Amyloid-β plaques, tau tangles [37,90]	E/I imbalance, PVI/PNN disruption [91,92]
**Key Brain Regions**	Hippocampus, cortex [93,94]	Prefrontal cortex, hippocampus [91]
**Autophagy-Related Genes**	↓ BECN1, ↓ ULK2, ↓ ATG5 [85,95]	↓ BECN1, ↓ ULK2, ↑ BCL2, ↑ ADNP [85]
**Protein Aggregates**	Extracellular Aβ, intracellular tau [94,96]	Synaptic/scaffolding protein buildup [85]
**Neuroinflammation**	Microglial activation, RAGE/NF-κB signaling [97]	MMP9/RAGE pathway, oxidative stress [63,92]
**Cognitive Symptoms**	Memory loss, executive dysfunction [98]	Working memory deficits, disorganized thinking [99]

### 2.4. Divergent Cell Death Pathways in Schizophrenia and AD: From Apoptosis to Necroptosis

The mechanisms of neuronal cell death are fundamental in the differentiation of neurodevelopmental disorders from neurodegenerative conditions. Cell death was always broadly categorized into apoptosis and necrosis; however, recent research has identified necroptosis as a distinct, regulated form of necrosis with significant relevance in brain pathology [100,101]. Apoptosis is a programmed, caspase-dependent process characterized by cell shrinkage and the formation of a non-inflammatory apoptotic body [101]. In contrast, necrosis is an unregulated, passive process resulting from acute injury, leading to cell swelling, membrane rupture, and a strong inflammatory response [100]. Necroptosis bridges these two concepts and presents as a form of programmed cell death, caspase-independent and reliant on the receptor-interacting protein kinase 1, receptor-interacting protein kinase 3, and mixed lineage kinase domain-like pseudokinase RIPK1/RIPK3/MLKL signaling pathway, culminating in membrane rupture and the release of DAMPs [100,102].

In post-mortem studies of schizophrenia, the classical neuropathological findings do not include evidence of widespread necrosis or the associated fibrillary gliosis, which has been a longstanding argument against its classification as a classic neurodegenerative disorder [103]. While some studies suggest that excessive neuronal apoptosis may contribute to the subtle, progressive gray matter changes observed in schizophrenia, this process does not typically induce the inflammation characteristic of necrosis [104]. Conversely, AD is defined by massive neuronal loss, where necroptosis and necrosis are central pathological hallmarks [105,106]. While apoptosis occurs in AD, it is not the predominant cell death modality. Instead, neurons frequently exhibit features of necroptosis, such as cellular swelling and DNA fragmentation [105]. The aberrant accumulation of Aβ and hyperphosphorylated tau directly triggers the necroptotic pathway, with elevated levels of key mediators like RIPK1 correlating with disease severity and neuroinflammation [102,105]. Unlike the immune-silent nature of apoptosis, the necroptosis observed in AD actively drives neuroinflammation, creating a vicious cycle that perpetuates neuronal injury and contributes significantly to the disease’s progressive neurodegenerative course [102,105,106].

## 3. Biomarkers and Diagnostic Tools

### 3.1. S100B in Neurodegeneration and RAGE Interaction

S100B is a calcium-binding protein predominantly produced by astrocytes in the CNS. It serves as a biomarker for various neurological conditions and plays a significant role in neurodegenerative processes. The review by Michetti et al. synthesizes findings from in vitro studies using cultured cells, in vivo experiments on animal models, and clinical analyses from human samples to reveal that S100B, primarily secreted by astrocytes, exhibits a concentration-dependent effect: it is neurotrophic at low levels, while at pathologically high concentrations, it becomes detrimental, promoting neuroinflammation and contributing to neuronal damage [107]. Elevated levels in cerebrospinal fluid (CSF) and/or blood serum are indicative of CNS injury or disease. For instance, increased serum S100B is associated with astroglial activation and potential BBB disruption. However, to date, the exact mechanism by which S100B crosses the BBB and enters the systemic circulation remains under investigation [108]. Studies using in vitro cell cultures, animal models (including S100B knockout mice), and clinical human data suggest that S100B influences BBB integrity by modulating tight junction proteins, which are critical for maintaining BBB permeability. 

Additionally, S100B, as shown in Figure 2, affects the expression of aquaporin-4, a key protein involved in the glymphatic system responsible for clearing extracellular proteins from the central nervous system [108]. S100B binds to RAGE, initiating signaling cascades that ultimately lead to inflammatory responses. This S100B/RAGE interaction activates pathways such as the MAPK pathway and nuclear factor kappa-light-chain-enhancer of activated B cells (NF-κB) translocation, resulting in the production of pro-inflammatory cytokines and contributing to neuroinflammation (Figure 2) [109]. In AD, the S100B/RAGE axis is of particular relevance since RAGE is upregulated in the AD brain and at the BBB, where its activation by S100B and other ligands triggers the generation of pro-inflammatory cytokines [110]. Zou et al. utilized rat models of traumatic brain injury, stretch-injured rat astrocytes, and rat brain microvascular endothelial cell cultures to demonstrate that S100B released after traumatic brain injury activates the RAGE receptor. This S100B/RAGE signaling enhances the expression, translocation, and activity of A disintegrin and metalloproteinase 17 (ADAM17), leading to the shedding of the endothelial glycocalyx, a critical component for vascular integrity. This, consequently, aggravates BBB dysfunction and increases vascular permeability, contributing to secondary brain injury [111]. Modulating the S100B-RAGE axis by using FPS-ZM1, a high-affinity RAGE-specific inhibitor, has shown a reduction in neuroinflammation and progressive improvement in cognitive functions in AD models. Specifically, AGEs-RAGE-activated rat models were intrahippocampally injected with AGEs and were successively treated with FPS-ZM1 intraperitoneal injections [112].

In schizophrenia, S100B is also increasingly recognized as a relevant biomarker and potential contributor to disease pathophysiology. Studies have reported elevated S100B levels in both serum and cerebrospinal fluid in patients with schizophrenia, reflecting glial pathology and potential BBB alterations [113,114]. Elevated S100B levels correlate with illness duration and the severity of positive symptoms, while their relationship with negative symptoms remains less consistent. Moreover, higher S100B levels have been associated with poorer cognitive performance, particularly in tasks involving verbal memory, speed processing, and executive function [113]. From a mechanistic point of view, S100B released by activated astrocytes binds to RAGE, triggering downstream inflammatory cascades involving NF-κB and MAPK pathways, which contribute to perpetual neuroinflammation and oxidative stress in schizophrenia, similar to mechanisms observed in AD [113,115]. The S100B/RAGE axis may also influence white matter integrity and oligodendrocyte function, potentially underlying connectivity deficits characteristic of schizophrenia [116]. The review by Tarasov et al. discusses evidence indicating that while antipsychotic treatments, such as haloperidol and clozapine, have shown in some studies on schizophrenia patients to reduce S100B expression, S100B levels often remain elevated in individuals with chronic schizophrenia. This persistence is particularly noted in patients exhibiting pronounced negative symptoms, despite the antipsychotic administration [115].

### 3.2. Cognitive Impairment in Alzheimer’s Disease and Schizophrenia

Understanding the affected cognitive domains, utilizing appropriate neuropsychological assessments for early diagnosis, and implementing strategies for cognitive rehabilitation are crucial components in managing AD [117]. Memory, language, attention, executive functions, and perceptual skills are all altered cognitive domains affected by AD. Short-term memory loss is commonly the earliest symptom, progressing to long-term memory loss together with difficulties in word-finding, naming objects, and reduced language fluency [117,118]. Early and accurate diagnosis of AD is essential for planned intervention and therapeutic strategies. Several neuropsychological assessments are used to evaluate cognitive function, such as the mini-mental state examination, a brief 30-point questionnaire assessing orientation, memory, attention, language, and visual and spatial skills [119]. The Alzheimer’s disease assessment scale cognitive subscale is a grading system specifically developed for AD, assessing memory, language, praxis, and orientation [120]. The most widely used test is the clock drawing test, which is a simple and quick assessment where individuals are asked to draw a clock showing a specific time, thus evaluating executive function, visuospatial abilities, and attention [121]. While there is no cure for AD itself, therapeutic strategies aim to maintain cognitive function and improve quality of life. Cognitive stimulation therapy, cognitive training, and social engagement, for instance, demonstrated that engaging patients in themed activities designed to enhance cognitive and social functioning can lead to significant improvements in cognitive function [122,123].

Cognitive impairment is also considered a central feature of schizophrenia, though the patterns of impairment differ substantially between the two disorders. In schizophrenia, deficits typically emerge early, sometimes even before the onset of psychosis, and remain relatively stable throughout life, affecting domains such as attention, executive function, processing speed, working memory, and even social cognition [124,125]. Unlike the progressive memory loss seen in AD, memory impairment in schizophrenia is usually less severe and often related to difficulties with encoding and retrieval rather than rapid forgetting [125]. Neuropsychological assessments tailored for schizophrenia, such as the Measurement and Treatment Research to Improve Cognition in Schizophrenia (MATRICS) Consensus Cognitive Battery and the Wisconsin Card Scoring Test, focus on these domains and are widely used in both research and clinical settings [126]. Importantly, cognitive remediation, psychological, and social interventions have shown moderate improvements in cognitive and functional outcomes for people with schizophrenia, emphasizing the need for early identification and targeted intervention [126,127,128]. Recent research also highlights that cognitive impairment in schizophrenia is largely resistant to standard antipsychotic treatment, underlining the importance of specialized cognitive approaches [124,125].

### 3.3. Psychiatric Manifestations as Early Indicators of Alzheimer’s Disease and Schizophrenia

In most cases, psychiatric manifestations precede noticeable cognitive symptoms and thus can serve as early indicators of the disease. Early psychiatric symptoms in AD may include mood disturbances such as depression and anxiety, personality changes, irritability, and social withdrawal [129]. Diagnosing AD based on psychiatric symptoms alone presents challenges due to their overlap with other psychiatric disorders. Comprehensive assessments, including neuropsychological testing and imaging, are essential for accurate diagnosis [119]. Management strategies encompass both pharmacological treatments, such as antidepressants and antipsychotics, and non-pharmacological interventions, including cognitive behavioral therapy and lifestyle modifications [123,129,130].

Similarly, in schizophrenia, early psychiatric and behavioral changes often precede the onset of classic psychotic symptoms. The prodromal phase of schizophrenia is frequently marked by mood disturbances, chronic anxiety, irritability, social withdrawal, and diminished emotional expression, features that can closely resemble those seen in the early stages of AD [131,132,133,134]. These non-specific symptoms may persist for months or years before the emergence of psychosis, making early diagnosis challenging and necessitating comprehensive clinical assessment [131,134]. Notably, anxiety and affective symptoms are increasingly recognized as important early features and potential intervention targets in schizophrenia, both for primary prevention and for reducing the risk of psychotic relapse [132]. Neuroinflammatory changes and neurocognitive deficits may also be present during this phase, reflecting the underlying brain alterations that precede full psychosis [133,135]. As in AD, early identification and intervention through a combination of pharmacological and psychosocial strategies are associated with improved long-term outcomes in schizophrenia [131,135].

Premorbid personality refers to the personality traits a person exhibited before any signs of the disorder appeared. In schizophrenia, premorbid personality is frequently characterized by traits within the schizophrenia spectrum, particularly schizoid personality disorder, schizotypal personality disorder, and avoidant personality features. These include social withdrawal, emotional flatness, introversion, poor sociability, and difficulties in psychosocial adjustment, often observable from childhood or adolescence. Studies show that the more pronounced these personality disorders are, the poorer the premorbid psychosocial adjustment, and these account for a significant portion of the variance in premorbid functioning among individuals who later develop schizophrenia [136,137]. In contrast, premorbid personality in AD is less defined by a specific spectrum but is often associated with increased neuroticism, introversion, dependence on others, restricted social activity, low frustration tolerance, and a need for daily guidance. There is also evidence that premorbid depressive traits and low levels of extraversion and conscientiousness are more common in those who later develop AD. All these features together may predispose individuals to noncognitive symptoms and behavioral disturbances as the disease progresses [138,139,140,141].

### 3.4. Differential Cognitive Impairment in Schizophrenia and Alzheimer’s: Psychiatric and Psychological Perspectives

In schizophrenia, the core psychiatric symptoms alter the way reality is interpreted and felt. The positive symptoms, such as auditory hallucinations and prominent delusions, are experiences added to a person’s reality and are associated with significant difficulties in daily functioning [142]. These are often accompanied by negative symptoms, with a profound lack of motivation, a flattening of emotional expression, reduced speech, and social withdrawal [99]. These are not character flaws but a core component of the schizophrenia syndrome, with over half of patients exhibiting at least one negative symptom [99]. Cognitive dysfunction is also a core feature, with moderate to severe deficits across domains like attention, memory, and executive functions [143]. Living with this altered reality inevitably gives rise to a powerful set of psychological symptoms. The constant presence of threatening voices in some patients or the oppressive belief that one is being watched generate intense and persistent fear, anxiety, and paranoia [142]. The struggle to think clearly or have coherent, real thoughts leads to frustration and typically to a debilitating loss of self-confidence [143]. Consequently, the social withdrawal seen in schizophrenia is not just a primary negative symptom, but it is also a psychological coping mechanism to reduce the overwhelming stimulation of social interaction and to hide the confusion of the illness from others [99,143].

In AD, the psychiatric symptoms are primarily the neuropsychiatric symptoms that result from the slow degradation of the brain [144]. While memory loss is the hallmark, nearly all people with AD develop neuropsychiatric symptoms during their disease. A meta-analysis found the most frequent neuropsychiatric symptom to be apathy, with a 49% prevalence, followed by depression, aggression, anxiety, and sleep disorders. Psychosis is also a major psychiatric symptom, with delusions and hallucinations being very common [145]. These are direct consequences of neurodegeneration and, as the disease progresses, symptoms like delusions and aggression become more frequent. The presence of neuropsychiatric symptoms is a prognostic marker, with psychosis being linked to a more rapid rate of cognitive decline [144]. From a psychological point of view, AD is often defined by the painful awareness of the progressive decline. In the early and mid stages, the individual is often extremely aware of their failing memory. This leads to a cascade of psychological reactions and reflections, such as experiencing a sense of loss of control, stress about declining abilities, and perceived guilt over the anticipated burden on caregivers [129]. The confusion and disorientation that stem from memory loss are not just cognitive errors but are highly terrifying experiences that can leave a person feeling lost and vulnerable even in their own home [129,144]. This loss of cognitive control directly enhances the loss of confidence and self-esteem, as the person can no longer trust their own perceptions and judgment abilities. These reactions, such as fear, grief, and frustration, are the emotional fallout of the underlying disease process [129,144,145].

### 3.5. Diagnostic Challenges and the Need for Comprehensive Evaluation in Schizophrenia and AD

The accurate diagnoses of schizophrenia and AD remain two of the most challenging tasks in neuropsychiatry, especially given the substantial overlap in symptoms between these disorders and a variety of organic brain diseases. Schizophrenia can be mimicked by numerous organic conditions, such as temporal lobe epilepsy, autoimmune encephalitis, neurosyphilis, brain tumors, and substance-induced psychosis, making a thorough diagnostic profile essential before referring to this condition [146,147]. The recent literature underscores the critical importance of comprehensive diagnostic workups, including brain MRI, electroencephalogram (EEG), lumbar puncture, neuropsychological assessment, drug urinalysis, and full blood work before setting definitive diagnoses such as schizophrenia or AD [146,147,148]. Performing brain imaging in the setting of both conditions is of extreme importance because clear differences can be seen and identified in order to go towards the correct diagnostic pathway. For instance, brain imaging in AD reveals pronounced and progressive atrophy, especially in the medial and temporal lobes, including the hippocampus, with eventual involvement of the entire cortex except for sensorimotor regions. Ventricular enlargement and widened sulci are also prominent in AD [149]. In schizophrenia, brain imaging shows more subtle and regionally specific changes, such as reduced gray matter and cortical thickness, particularly in the frontal, anterior cingulate, and temporal cortices, as well as the hippocampus [150]. These deficits are different from those presenting in AD and do not show the same pattern of cortical atrophy or ventricular enlargement. In the absence of such evaluations, patients should be classified as experiencing psychosis not otherwise specified (NOS), thus avoiding the contamination of research cohorts with heterogeneous and potentially treatable organic conditions. This approach is supported by evidence that pseudopsychiatric presentations are common in clinical practice and that the exclusion of organic causes is essential for both patient management and the integrity of scientific work [151,152,153].

### 3.6. Lumipulse G pTau217/ß-Amyloid 1-42 Plasma Ratio

The Lumipulse G phosphorylated tau protein at amino acid position 217 (pTau217)/amyloid beta protein fragment 1-42 (Aß1-42) plasma ratio is a recently FDA-cleared blood test designed to aid in the diagnosis of AD by detecting amyloid pathology in the brain. This test measures the levels of pTau217 and Aß1-42 in plasma and calculates their ratio, which has shown a strong correlation with the presence of amyloid plaques, a hallmark of AD. Several peer-reviewed studies have validated the clinical utility and accuracy of this test. For instance, a study published in 2025 evaluated this ratio using both amyloid and tau positron emission tomography (PET) imaging as reference standards in clinical and community cohorts. The values obtained from this ratio, evaluated using PET imaging, showed clinically equivalent results when compared to established CSF biomarkers, thus achieving high diagnostic accuracy [154]. Another study demonstrated that plasma pTau217 and the pTau217/Aβ42 ratio, measured on the automated Lumipulse platform, strongly correlated with CSF total tau. The ratio showed excellent accuracy, sensitivity (91.8%), and specificity (95.1%) for distinguishing AD pathology, surpassing other blood-based biomarkers [155]. These conducted studies, in addition to the recent FDA clearance, make the Lumipulse G pTau217/ß-Amyloid 1-42 Plasma Ratio a clinically validated, non-invasive blood test with high diagnostic accuracy for detecting amyloid pathology in AD.

## 4. Therapeutic Strategies

### 4.1. Current Treatments: Antipsychotics, Metformin, and Cholinesterase Inhibitors

Currently available treatments for AD and schizophrenia primarily focus on symptomatic relief rather than halting the condition itself. Cholinesterase inhibitors, such as galantamine, rivastigmine, and donepezil, are approved for mild to moderate AD. They work by enhancing cholinergic transmission through the inhibition of acetylcholinesterase, which is an enzyme that degrades acetylcholine, an essential neurotransmitter involved in learning and memory [156,157]. As a medication used to slow the symptomatic progression of moderate-to-severe AD, memantine holds a key position in neuropharmacology [158,159]. Memantine’s efficacy is based on the glutamate excitotoxicity principle, by which the excessive stimulation of N-methyl-D-aspartate receptors (NMDARs) by the neurotransmitter glutamate contributes to the neuronal damage seen in AD [158,159]. Memantine functions as a low-to-moderate affinity, voltage-dependent, uncompetitive antagonist of NMDARs. Thus, by selectively inhibiting pathological glutamate activity and reducing excessive calcium influx, it has been shown in clinical trials that the drug reduces the rate of clinical deterioration in patients with moderate-to-severe AD [159,160]. In schizophrenia, antipsychotic medications, spanning over three generations, target dopamine D2 receptors to alleviate positive symptoms such as hallucinations [161]. These include haloperidol, a first-generation typical antipsychotic that acts as a potent D2 antagonist; clozapine, a second-generation atypical antipsychotic with a broader receptor profile, including serotonin 5-hydroxytryptamine 2A (5-HT2A) receptor antagonism; and aripiprazole, a third-generation medication working as a D2 partial agonist [161,162,163]. While drugs such as clozapine are effective in the treatment of psychosis, they are also well-known to induce significant metabolic side effects, such as weight gain, insulin resistance, and dyslipidemia, collectively known as metabolic syndrome [164,165]. This iatrogenic metabolic syndrome is a major independent risk factor for developing AD. Recent pharmaco-epidemiological studies have confirmed this link, suggesting that the prescription of antipsychotics like clozapine is significantly associated with an increased risk of AD [164]. The proposed mechanism involves the suppression of microglial-mediated Aβ clearance, leading to Aβ accumulation in the brain, thus creating a challenging clinical scenario where the treatment of one disorder directly elevates the risk for another, complicating long-term patient management. Conversely, attempts to modulate the glutamatergic system, a key target in AD, can precipitate severe psychotic symptoms. While memantine is a moderate-affinity NMDAR antagonist used in AD, more potent NMDAR antagonists like ketamine and phencyclidine are known to induce profound, transient psychoses that mimic both the positive and negative symptoms of schizophrenia [166,167]. In fact, the ketamine model is considered one of the best pharmacological models for studying schizophrenia in healthy volunteers, as it reliably produces schizophrenia-like symptoms by blocking NMDARs [168,169]. Metformin has become a critical adjunctive therapy in the management of schizophrenia, specifically to counteract the iatrogenic metabolic syndrome frequently caused by second-generation antipsychotics [165]. The importance of metformin is underscored by robust evidence and emerging clinical guidelines that recommend its use for both the prevention and treatment of antipsychotic-induced weight gain (AIWG) [170]. For prevention, the new evidence-based guidelines suggest that co-starting metformin treatment when initiating a high-risk antipsychotic is the most effective strategy. This approach can reduce weight gain by an average of 4 kg compared to controls, highlighting that preventing weight gain is a more effective strategy than attempting to reverse it later [170]. However, recent research has shifted towards addressing cognitive and negative symptoms, which are often resistant to traditional therapies [33].

### 4.2. Beyond Genetics: The Role of Environment and Epigenetics

All the iatrogenic links emphasize that an exclusive focus on genetics is inadequate for explaining the etiology and course of either schizophrenia or AD. Environmental factors play a substantial and undeniable role. The development of schizophrenia is multifactorial, reflecting a complex interaction between genetic vulnerabilities and environmental contributors [171]. Established environmental risk factors include obstetric complications, urban living, migration, childhood trauma, and cannabis use, all of which influence an individual’s likelihood of developing the disorder [172,173]. Similarly, the progression of AD is not solely dictated by genetics. Environmental factors and lifestyle contribute significantly, and their influence is often mediated through epigenetics, molecular modifications like deoxyribonucleic acid (DNA) methylation, and histone modifications that alter gene expression without changing the DNA sequence itself [174]. This concept emphasizes the fact that genetic predispositions often require an environmental trigger to manifest as a clinical disorder. Therefore, a comprehensive understanding of both schizophrenia and AD necessitates a holistic view that comprehends genetic vulnerabilities, iatrogenic risks, and diverse environmental exposures.

### 4.3. Emerging Targets: mTOR, SYK, and S100B-RAGE

Understanding molecular mechanisms of both disorders has opened avenues for targeting key signaling pathways. The mTOR pathway has emerged as a central node linking autophagy, metabolism, and cell survival. The inhibition of mTORC1 has been shown to restore autophagy and promote the clearance of Aβ and tau aggregates in transgenic mouse models [46,47]. Similarly, mTOR-related signaling appears dysregulated in schizophrenia, further supporting its role as a shared therapeutic target [46,48,49]. SYK is a non-receptor tyrosine kinase that regulates microglial response to Aβ aggregates via phagocytosis. The overactivation of SYK can lead to chronic neuroinflammation, while its inhibition impairs Aβ clearance, suggesting the need for finely tuned modulatory strategies (Figure 3) [43,44]. RAGE, a transmembrane receptor implicated in chronic inflammation, facilitates the influx of Aβ across the BBB and amplifies cytokine release via NF-κB signaling. RAGE inhibitors like FPS-ZM1 and Azeliragon have shown promise in reducing amyloid deposition and improving cognitive outcomes in transgenic mouse models of AD and in clinical trials [62]

Increasingly, the S100B/RAGE axis is recognized as both a mechanistic driver and a promising biomarker in AD and schizophrenia. The binding of S100B to RAGE activates NF-κB and MAPK pathways, leading to the release of pro-inflammatory cytokines and further promoting amyloid accumulation and neuronal injury. This interaction not only exacerbates amyloid pathology in AD but also contributes to chronic inflammation and BBB dysfunction, highlighting the potential therapeutic value of targeting the S100B/RAGE axis in neurodegenerative disorders [109,110]. Similarly, in schizophrenia, elevated S100B levels have been consistently reported in serum and cerebrospinal fluid, reflecting glial dysfunction and ongoing neuroinflammation. The S100B/RAGE interaction is thought to mediate inflammatory signaling in the brain, contributing to white matter abnormalities, oligodendrocyte loss, and cognitive impairment observed in schizophrenia. While S100B is not specific to schizophrenia, its persistent elevation underscores its value as a peripheral biomarker for glial dysfunction and disorder activity. Studies also suggest that S100B levels may correlate with symptom severity and treatment response, although specificity remains limited due to its elevation in other neuropsychiatric and neurodegenerative conditions [109,175,176].

### 4.4. The Roles of Psychiatry and Psychology in the Context of Schizophrenia and Alzheimer’s Management

For a neurodevelopmental disorder like schizophrenia, psychology and psychiatry offer distinct but highly complementary roles in patient care, while for a progressive neurodegenerative disorder such as AD, the collaboration between the two fields is equally essential. In schizophrenia, from a psychiatrist’s perspective, the primary role involves managing the illness with antipsychotic medications [125,177]. These drugs are effective for the positive symptoms like hallucinations and delusions because they act on neurotransmitter systems, such as the dopamine D2 receptor [125]. However, these agents have little to no impact on the core cognitive impairments of the disorder [125]. Another aspect on which a psychiatrist focuses is the biological basis of cerebral dysfunction [178]. Here, there is a high responsibility for setting differential diagnoses to rule out other conditions and for managing the patient’s overall medical needs, which often involve a combination of pharmacological and psychosocial approaches [177]. A psychologist, instead, focuses on the cognitive and behavioral manifestations of the illness, which are considered core features distinct from positive and negative symptoms [125]. Neuropsychologists use comprehensive cognitive assessments to map a patient’s specific profile of impairment [179]. This step is critical for distinguishing the cognitive patterns of schizophrenia from other disorders like AD and for supporting the development of targeted treatment interventions [179]. For this condition, there is a need to provide evidence-based psychological therapies as an adjunct to pharmacotherapy [180]. This includes cognitive remediation, a set of behavioral training exercises that produces short-term improvements in cognitive functioning. It also includes therapies like social skills training to improve daily functioning. These psychosocial interventions are considered a vital part of a multimodal treatment model [177,180]. In the context of AD, psychiatrists prescribe and monitor the so-called antidementia compounds to manage cognitive symptoms, monitoring and managing also behavioral and psychological symptoms arising during the course of the disease [181]. A crucial psychiatric role is treating the psychosis (delusions and hallucinations) that affects up to 50% of AD patients [182]. This is typically managed with antipsychotic medications, though their efficacy can be modest in this population. In psychology, detailed cognitive testing is a cornerstone for an AD diagnosis [179]. Neuropsychological assessment can identify the hallmark memory profile of AD: impaired learning combined with poor recall of newly learned verbal information. This helps differentiate it from the cognitive profile seen in conditions like schizophrenia [179]. Psychology emphasizes non-pharmacological interventions to improve patients’ quality of life, including cognitive behavioral therapy, which has been shown to be effective in helping patients develop disease management skills [183].

### 4.5. Translational Outlook and Personalized Medicine

Despite the progress in understanding neurodegenerative and psychiatric disorders, translating findings into clinical practice remains a challenge. Heterogeneity in genetic profiles, symptom manifestation, and disorder progression necessitates a move towards personalized medicine. The integration of genomic data, biomarker profiles, and AI-based predictive tools could optimize diagnosis, monitor disorder progression, and guide treatment choice. For instance, individuals with RAGE polymorphism or altered S100B levels may benefit from RAGE-targeted therapies, while patients with mTOR pathway mutation could benefit from mTOR inhibition or autophagic modulation. For example, rapamycin, an mTOR inhibitor, has been shown in multiple AD mouse models to restore autophagy, reduce Aβ and tau accumulation, and improve cognitive function by inhibiting mTORC1 activity [184,185,186,187]. These effects are particularly relevant for patients with evidence of mTOR pathway overactivation or genetic mutations affecting autophagic processes. Similarly, metformin, widely used for type 2 diabetes, activates AMP-activated protein kinase (AMPK) and enhances autophagy, with studies demonstrating reduced Aβ pathology, anti-inflammatory effects, and improved cognitive performance in preclinical models of AD [188]. Metformin’s dual action on metabolic and neurodegenerative pathways makes it especially promising for individuals with insulin resistance or metabolic syndrome alongside cognitive decline.

Personalized medicine in schizophrenia is moving toward matching treatments to an individual’s genetic, biomarker, and clinical profile, aiming to improve outcomes beyond the traditional trial-and-error approach [189,190,191,192]. For example, patients with elevated inflammatory markers such as IL-6 or C-Reactive Protein (CRP) may benefit from adjunctive anti-inflammatory agents like minocycline or celecoxib, as these subgroups show greater immune dysregulation and may respond better to immunomodulatory treatments [193,194]. Meanwhile, individuals with rare mutations in genes such as GRIN2A, which encodes a subunit of the NMDA receptor and is associated with glutamatergic dysfunction, could be candidates for therapies targeting glutamate pathways, including NMDA receptor modulators or novel agents currently in development (Figure 3) [195,196,197].

## 5. Conclusions, Research Gaps, and Future Directions

AD remains a complex neurodegenerative disorder characterized by cognitive impairment, neuroinflammation, BBB dysfunction, and psychiatric comorbidities. The involvement of molecular pathways such as SYK, mTOR, and the RAGE/S100B axis highlights promising therapeutic targets for intervention. Moreover, psychiatric disorders, particularly depression, anxiety, and schizophrenia, show notable overlaps and common pathophysiological mechanisms with AD, underscoring their significance as both comorbidities and potential early indicators of disease onset. Despite considerable research efforts, several critical gaps remain. These include the heterogeneity of disease phenotypes, the absence of reliable biomarkers for early diagnosis, and the incomplete understanding of the molecular crosstalk between inflammatory, neurodegenerative, and psychiatric pathways. Furthermore, many therapeutic strategies have failed in clinical trials, particularly those targeting amyloid deposition, tau pathology, and chronic neuroinflammation, reflecting the need for tailored approaches. Priorities include developing early-stage biomarkers for both psychiatric symptoms and neuroinflammatory processes, investigating combination therapies that concurrently target multiple molecular pathways, and enhancing neuroimaging techniques capable of detecting subtle BBB changes and inflammatory markers during the preclinical stages.

## Figures and Tables

**Figure 1 ijms-26-06237-f001:**
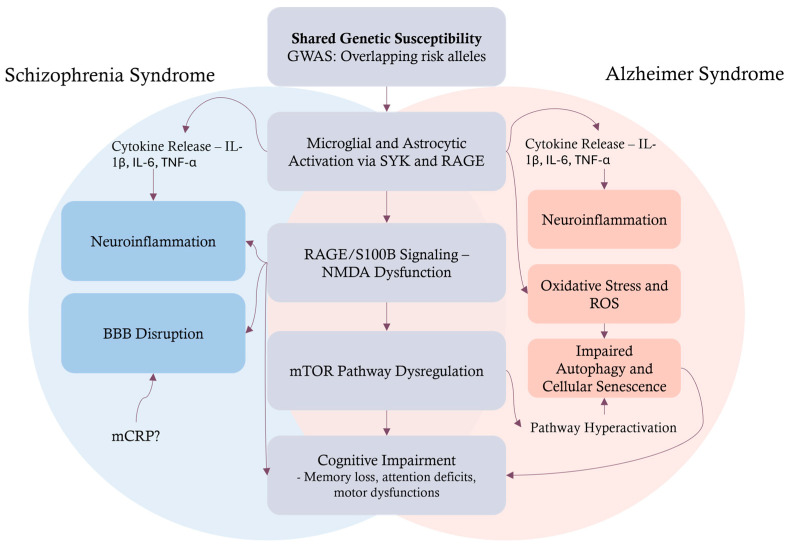
Overlapping and distinct pathophysiological mechanisms in schizophrenia and Alzheimer’s disease leading to cognitive impairment. This schematic outlines a model detailing how schizophrenia (left, blue-shaded area) and Alzheimer’s disease (right, red-shaded area) develop cognitive impairment through both unique and shared pathological pathways. The model begins with a common foundation of shared genetic susceptibility, identified by overlapping risk alleles from GWAS, which is known to trigger microglial and astrocytic activation. This activation, involving signaling through SYK and RAGE, proceeds with a central pathogenic cascade. Key elements of the cascade include dysregulated RAGE/S100B signaling, subsequent NMDA receptor dysfunction, and, ultimately, mTOR pathway dysregulation, all converging to produce cognitive deficits like memory loss and attention problems. Distinctly, the pathways predominantly associated with Schizophrenia emphasize cytokine release (IL-1β, IL-6, TNF-α), leading to pronounced neuroinflammation and BBB disruption. These schizophrenia-specific processes are shown to significantly impact and feed into the shared central mechanism, particularly exacerbating RAGE/S100B signaling and NMDA dysfunction. Conversely, while AD also involves cytokine release, it distinctively leads to heightened oxidative stress and ROS production, which in turn contribute to impaired autophagy and cellular senescence. These AD-specific dysfunctions are illustrated as directly influencing the shared mTOR pathway dysregulation and contributing to cognitive decline, with a feedback loop also suggesting a direct impact on cognitive impairment. The diagram thus highlights how divergent initial pathologies can converge on common molecular pathways, illustrating a complex interplay that ultimately leads to the cognitive symptoms observed in both disorders. GWAS: genome-wide association studies; BBB: blood–brain barrier; SYK: spleen tyrosine kinase; RAGE: receptor for advanced glycation end-products; S100B: S100 calcium-binding protein B; NMDA: N-methyl-D-aspartate; AD: Alzheimer’s disease; ROS: reactive oxygen species; mTOR: mechanistic target of rapamycin; IL-1β: interleukin 1-beta; IL-6: interleukin 6; TNF-α: tumor necrosis factor alpha; mCRP: monomeric C-reactive protein.

**Figure 2 ijms-26-06237-f002:**
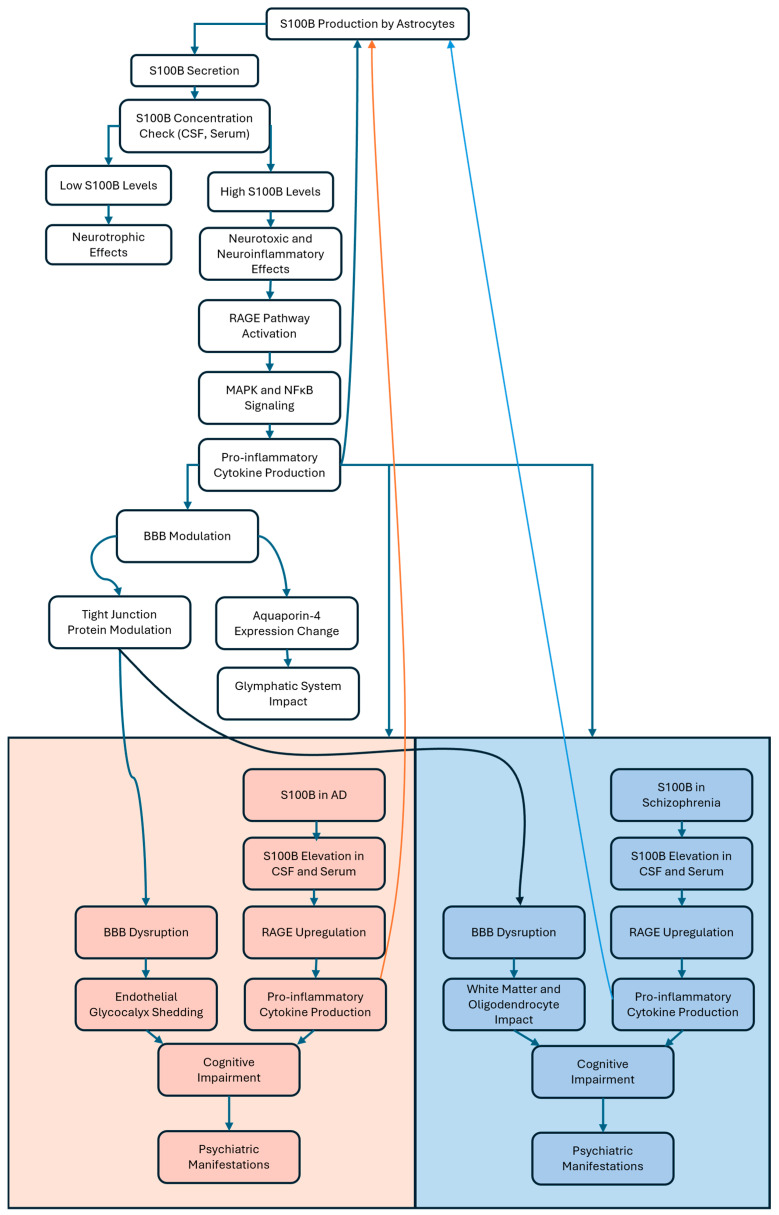
Diagram illustrating the role of the S100B protein, primarily produced and secreted by astrocytes, in neurological diseases, focusing on implications in AD and schizophrenia. The concentration of S100B rules the effects: low concentrations are neurotrophic, while high concentrations initiate neurotoxic and neuroinflammatory cascades. The neurotoxic pathway involves the activation of the RAGE receptor, subsequent MAPK and NFκB signaling, and, ultimately, the production of pro-inflammatory cytokines. These cytokines modulate the BBB by altering tight junction proteins and aquaporin-4 expression, impacting the glymphatic system and leading to BBB disruption. The diagram then describes how high concentrations of S100B and BBB dysfunction contribute to AD (pink area and boxes) and schizophrenia (blue area and boxes). In both conditions, elevated S100B levels in CSF and serum lead to RAGE upregulation and further pro-inflammatory cytokine production, leading to cognitive impairment and psychiatric manifestations. Specific to AD, endothelial glycocalyx shedding contributes to BBB disruption. In schizophrenia, BBB disruption is linked to white matter and oligodendrocytes, with a feedback loop potentially enhancing further S100B production. Both pathways lead to neuropsychological assessments and therapeutic interventions. S100B: S100 calcium protein-B; CSF: cerebrospinal fluid; RAGE: receptor for advanced glycation end-products; MAPK: mitogen-activated protein kinase; NF-κB: nuclear factor-kappa B; BBB: blood–brain barrier; AD: Alzheimer’s disease.

**Figure 3 ijms-26-06237-f003:**
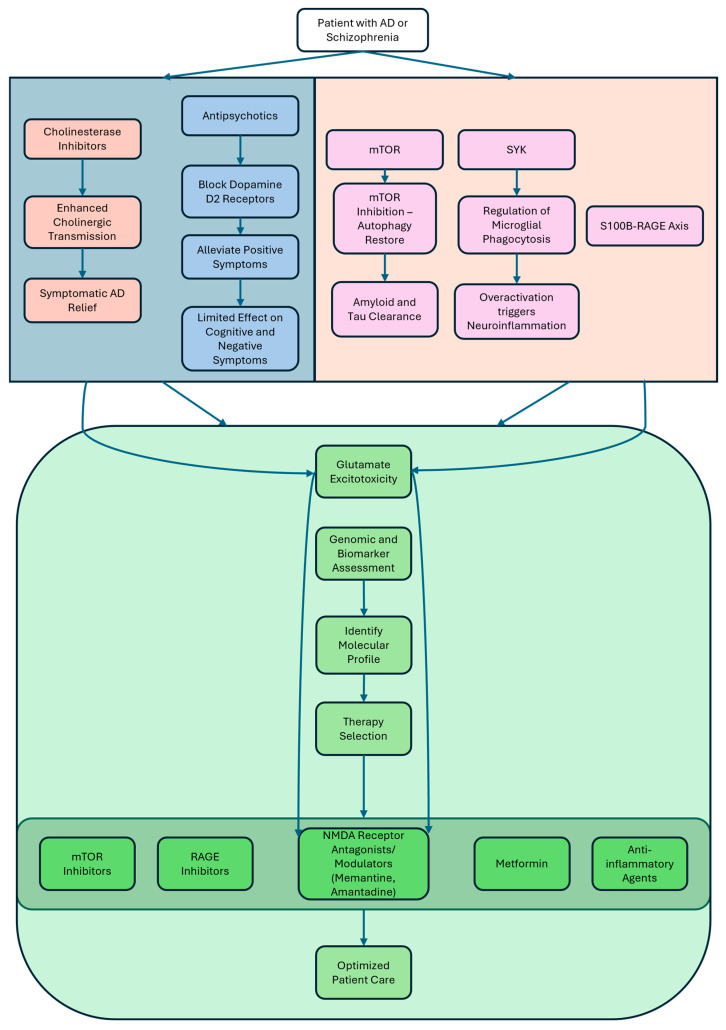
Schematic overview of current and emerging therapeutic strategies for patients with AD or schizophrenia, emphasizing the role of personalized medicine. The diagram is divided into two main sections. The upper section (blue and peach boxes) summarizes established and experimental treatments for AD and schizophrenia. On the left (blue), cholinesterase inhibitors are shown to enhance cholinergic transmission and provide symptomatic relief in AD, while antipsychotics block dopamine D2 receptors to alleviate positive symptoms in schizophrenia, with limited efficacy for cognitive and negative symptoms. The right side (peach) highlights emerging molecular targets, including mTOR and SYK pathways. mTOR inhibition aims to restore autophagy and promote amyloid and tau clearance, while SYK regulates microglial phagocytosis; its overactivation, especially via the S100B-RAGE axis, can trigger neuroinflammation. The lower section (green) illustrates a personalized medicine approach. It begins with consideration of glutamate excitotoxicity, which refers to neuronal damage caused by the excessive activation of glutamate receptors, primarily NMDARs, leading to pathological calcium influx, oxidative stress, and cell death. This is a central mechanism involved in neurodegeneration in AD and contributes to synaptic dysfunction in Schizophrenia. NMDAR antagonists and modulators are agents that selectively inhibit the activity of NMDARs, which are critical for synaptic plasticity, learning, and memory, but are involved in excitotoxic neuronal injury when excessively activated. Genomic and biomarker assessment is performed to identify the patient’s molecular profile, guiding therapy selection. Potential targeted treatments include mTOR inhibitors, RAGE inhibitors, metformin, and anti-inflammatory agents. The goal is to optimize patient care by tailoring interventions to individual molecular characteristics.

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
