# Peer review of "The Inflammatory Nexus: Unraveling Shared Pathways and Promising Treatments in Alzheimer’s Disease and Schizophrenia"

_ijms, 2025, doi:10.3390/ijms26136237_

Round 1
Reviewer 1 Report
Comments and Suggestions for Authors
AUTHORS
Please, first author should use institutional e-mail address, instead of gmail.
ABSTRACT
Please, explain all acronyms, such as: “S100B”, “MMP9”.
KEYWORDS
Please capitalize “Alzheimer” and explain acronyms “S100B and “RAGE”.
TEXT
Please, explain all acronyms, such as: “PSEN1”, “PSEN2”, “APP”, “LRRK2”, “SNCA”, “FDA”, “MRI”, “5xFAD”, “IL-6”, “CCL2”, “CXCL10”, “mTORC1”, “mTORC2”, “FPS-ZM1”, “MMP9”, “S100”, “ICAM”, “ULK2”, “BECN1”, “ADAM17”, “MATRICS”, “nr.”, etc.
Please, all non-English words should be in italics, such as “et al”, “nucleus basalis”, “in vivo”, “post mortem”, “studii universitare de doctorat”, etc.
Please, delete all text regarding bipolar disorder. That is not necessary for this article.
Please, avoid using stigmatizing words like “schizophrenic”. Use instead “patients with schizophrenia”.
Whenever using the acronym MRI the authors should specify the part of the body that was studied, eg “brain MRI”.
When citing reference 16 the Authors cannot use the word “disease”. Schizophrenia is not a disease. Officially it is a psychiatric disorder. Schizophrenia is also a pseudogenetic syndrome. All psychiatric conditions increase the risk of dementia in later life. Genetics is not as important as previously thought. The risk of psychiatric patients die with Alzheimer is much more related to environment (nurture) than genetics (nature), eg smoking, alcohol, drugs, obesity, dyslipidemia, diabetes mellitus, arterial hypertension, metabolic syndrome, mortification, etc.
Wow! “Type 3 diabetes”? Interesting. Are the authors talking about diabetes mellitus or diabetes insipida? Please, clarify, again when writing about “type 2 diabetes”.
Please, acknowledge undiagnosed organic psychoses as cofounders in the diagnosis of Alzheimer and schizophrenia. Patients who were never studied with brain magnetic resonance imaging, electroencephalogram, lumbar puncture, neuropsychological assessment, drug urinalysis or complete blood work should never be labeled with Alzheimer or schizophrenia, but with psychosis non other specified. Schizophrenia is the greatest imitated syndrome of neuropsychiatry. Beware of pseudoschizophrenia. Please, avoid the mistake repeated ad nauseam in the twentieth century when millions of patients with unspecified psychoses were pooled together in studies of schizophrenia, without necessary ruling out of organic causes. The sample of patients should not be contaminated with any kind of organic condition that can mimic schizophrenia, namely brain anomaly, temporal lobe epilepsy, encephalitis, cannabis abuse, syphilis, etc.
Please acknowledge the differences between neuronal apoptosis, necropoptosis and necrosis. Classically in schizophrenia autopsies there were no signs of necrosis. On the other hand, neuronal necrosis is a classic hallmark of Alzheimer’s syndrome.
THERAPEUTIC STRATEGIES
Please, add memantine to medication used to slow the progression of moderate-to-severe Alzheimer’ syndrome. And explain, it simple class mechanism, eg low-affinity voltage-dependent uncompetitive antagonist at glutamatergic NMDA receptors.
Please, delete “olanzapine” and “risperidone”. Add instead “clozapine” and “haloperidol”. Then explain that there are three generations for antipsychotic medication, eg haloperidol (first generation), clozapine (second generation” and aripipirazole (third generation).
Please, explain the importance of metformin as it is already recommended by guidelines in the treatment of schizophrenia, aiming the prevention / treatment of iatrogenic metabolic syndrome (mainly diabetes mellitus type 2 and obesity secondary to second generation anti-psychotics such as clozapine).
Please explain the iatrogenic effects of medication in both disorders: eg second generation antipsychotics (clozapine, etc) used in schizophrenia increase the risk factor for Alzheimer’s syndrome, such as metabolic syndrome. On the other hand, some NMDAr interfering molecules are dangerous and can induce sever schizophrenia-like psychoses (eg ketamine and phencyclidine). Caution is needed when the Authors try to explain everything from the point of view of genetics, while environment and much influence in the course of both syndromes.
FIGURES
Figure 1. I would use “schizophrenia syndrome” and “Alzheimer syndrome” instead of disease.
Figure 2. Delete the boxes with “Neuropsychological Assessments” and “Therapeutic interventions”. Those make no sense in a schema regarding “neuroinflammatory cascades”
Figure 3. Please, review your English: instead of “negative symproms” it should be negative “symptoms”. Again the Figure lacks logic. It would be much more interesting to include here the anti-NMDAr molecules (eg memantine) because glutamate toxicity is the biggest common issue in Alzheimer and schizophrenia syndrome’s neuropathology. One more thing: there are studies showing that clearing beta amyloid and tau protein is not enough to change the prognosis of Alzheimer’s syndrome. It seems that those biomarkers might be just an epiphenomenon of the neuropathology of the disease: eg either consequence and not cause of neuronal degeneration.
TABLES
Table 1. Figure 1. I would use “schizophrenia syndrome” and “Alzheimer syndrome” instead of disease. Please explain all used acronyms at the bottom of the Table.
Author Response
Please see the attached file we thank the reviewer for their constructive comments that have enabled us to improve the manuscript considerably

Reviewer 2 Report
Comments and Suggestions for Authors
This review discusses the overlap in the molecular and neuroinflammatory pathomechanism between schizophrenia (SCZ) and Alzheimer's disease (AD), two represenative psychiatric and neurodegenerative disorders. The common signal transduction pathway and the common process of pathophysiology in SCZ ans AD is described in detail with complete references. The common signaling pathways--- spleen tyrosine kinase (SYK); the S100B/receptor for advanced glycation end-product (RAGE) axis; MAPK and NF-kB signaling; mammalian target of rapamycin (mTOR)---produce common pathophysiology: neuroinflamamtion; oxidative stress and ROS; autophagy dysfunction; blood brain barrier (BBB) disruption; and cognitive impairment. S100B/RAGE axis in astrocytes is concluded to be of common impotance between SCZ and AD.
The present Reviewer would suggest the following points for miner revision.
| SCZ and AD have been traditionally considered to be different brain diseases, psychiatric and neurological diseases. SCZ gets sick in young, ~10-20 years old, and AD in old ages, ~50-60 years old; AD is a typical neurodegenerative disease with neuronal death, but SCZ does not reveal marked neuronal death; imaging data are different between SCZ and AD. These differences should be also discussed.
| Table 1 could become more useful by includeing references on each subject.
| Fig 1. Neuroinflammation should be shown also in AD .
| In psychology and psychiatry, symptoms of cognitive impairment in SCZ and AD are considered to be somewhat different. This should be discussed.
| GWAS shows differences in susceptobility genes between SCZ and AD.
| Premorbid personalities in SCZ and AD are considered to be different.
| Miner typographical error. line 244. Fig 1 regend: mTOR: mechanisitc Target of Rapamycin
Author Response

(The authors gave the same response as above.)
